# Effects of Manganese Porphyrins on Cellular Sulfur Metabolism

**DOI:** 10.3390/molecules25040980

**Published:** 2020-02-22

**Authors:** Kenneth R. Olson, Yan Gao, Andrea K. Steiger, Michael D. Pluth, Charles R. Tessier, Troy A. Markel, David Boone, Robert V. Stahelin, Ines Batinic-Haberle, Karl D. Straubg

**Affiliations:** 1Indiana University School of Medicine-South Bend Center, South Bend, IN 46617, USA; yangao@iu.edu (Y.G.); crtessie@iu.edu (C.R.T.); daboone@iu.edu (D.B.); 2Department of Biological Sciences, University of Notre Dame, Notre Dame, IN 46556, USA; 3Department of Chemistry and Biochemistry, University of Oregon, Eugene, OR 97403, USA; asteiger@uoregon.edu (A.K.S.); pluth@uoregon.edu (M.D.P.); 4Indiana University School of Medicine, Riley Hospital for Children at IU Health, 705 Riley Hospital Dr, RI 2500, Indianapolis, IN 46202, USA; tmarkel@iupui.edu; 5Department of Medicinal Chemistry and Molecular Pharmacology, Purdue University, West Lafayette, IN 47907, USA; rstaheli@purdue.edu; 6Department of Radiation Oncology, School of Medicine, Duke University, Durham, NC 27710, USA; ibatinic@duke.edu; 7Central Arkansas Veteran’s Healthcare System, Little Rock, AR 72205, USA; Karl.Straub@va.gov; 8Departments of Medicine and Biochemistry, University of Arkansas for Medical Sciences, Little Rock, AR 72202, USA

**Keywords:** reactive sulfide species, ROS, antioxidants, Mn porphyrins, SOD mimetics, H_2_S, polysulfides

## Abstract

Manganese porphyrins (MnPs), MnTE-2-PyP^5+^, MnTnHex-2-PyP^5+^ and MnTnBuOE-2-PyP^5+^, are superoxide dismutase (SOD) mimetics and form a redox cycle between O_2_ and reductants, including ascorbic acid, ultimately producing hydrogen peroxide (H_2_O_2_). We previously found that MnPs oxidize hydrogen sulfide (H_2_S) to polysulfides (PS; H_2_S_n_, n = 2–6) in buffer. Here, we examine the effects of MnPs for 24 h on H_2_S metabolism and PS production in HEK293, A549, HT29 and bone marrow derived stem cells (BMDSC) using H_2_S (AzMC, MeRho-AZ) and PS (SSP4) fluorophores. All MnPs decreased intracellular H_2_S production and increased intracellular PS. H_2_S metabolism and PS production were unaffected by cellular O_2_ (5% versus 21% O_2_), H_2_O_2_ or ascorbic acid. We observed with confocal microscopy that mitochondria are a major site of H_2_S production in HEK293 cells and that MnPs decrease mitochondrial H_2_S production and increase PS in what appeared to be nucleoli and cytosolic fibrillary elements. This supports a role for MnPs in the metabolism of H_2_S to PS, the latter serving as both short- and long-term antioxidants, and suggests that some of the biological effects of MnPs may be attributable to sulfur metabolism.

## 1. Introduction

Manganese-centered porphyrins (MnPs) have received increasing interest as therapeutic agents in the treatment of cancer and as cytoprotectants of normal tissue following radiation therapy. Three of these compounds, MnTE-2-PyP^5+^ (MnTE, AEOL10113, BMX-010), MnTnHex-2-PyP^5+^ (MnTnHex) and MnTnBuOE-2-PyP^5+^ (MnTnBuOE, BMX-001), appear to be especially efficacious in this regard and MnTBuOE-2-PyP^5+^ and MnTE-2-PyP^5+^ have progressed to clinical trials [1]. The pyridyl groups of these three MnPs have been uniquely modified to withdraw electrons from the manganese center, resulting in reduction potentials close to that of the endogenous antioxidant enzyme, superoxide dismutase (SOD; [1,2,3]).

While early research on these compounds focused on their SOD-mimetic and antioxidant attributes, recent evidence suggests that substantial biological activity is derived from their paradoxical oxidant activity through an increase in hydrogen peroxide (H_2_O_2_) in cells. This activates cytoprotective antioxidant defenses in healthy cells, presumably mediated to a large extent through the peroxidation of Keap-1 which frees it from Nrf2 allowing the latter to translocate to the nucleus and activate the antioxidant defenses. Peroxide also creates lethal oxidative stress in malignant cells [1,4]. However, unlike their ability to dismute superoxide, in this instance MnPs are believed to produce H_2_O_2_ by MnP cycling between one-electron reduction by a cellular reductant such as ascorbic acid (AA) and one-electron oxidation by oxygen. Ascorbic acid reduces Mn(III)P to Mn(II)P and oxygen reoxidizes it to Mn(III); this also reduces oxygen to superoxide. The superoxide produced from this reaction then dismutes, either spontaneously or catalyzed by SOD or MnPs, to produce O_2_ and H_2_O_2_. H_2_O_2_ then oxidizes regulatory protein cysteines thereby conferring the respective cytoprotective effects in healthy cells and cytotoxic reactions in malignant ones [1].

We have previously shown that both SOD1 and SOD2 and the above MnPs oxidize hydrogen sulfide (H_2_S) to form polysulfides in buffer [5,6]. In the present study we examine the effects of MnPs on endogenous sulfur metabolism in a variety of cell lines. We show that MnPs decrease the rate of endogenous H_2_S production and increase intracellular polysulfides in cells. These results suggest that MnPs oxidize intracellular H_2_S to form polysulfides, which is consistent with our previous observations of these reactions in buffer. We also observed that these reactions were independent of either ascorbic acid or hydrogen peroxide, suggesting that peroxide may not be involved in polysulfide production during H_2_S oxidation. Polysulfides formed by these reactions are potent antioxidants and can also initiate cytoprotective effects through persulfidation of Keap-1 and liberation of Nrf2. Collectively, these experiments support the hypothesis that MnPs affect intracellular sulfide metabolism and that the products from these reactions can directly contribute to the antioxidant effects of the MnPs.

## 2. Results

### 2.1. Effects of MnPs on Sulfur Metabolism in HEK293 Cells

The effects of MnTE, MnTnHex and MnTnBuOE on intracellular H_2_S (AzMC fluorescence) and polysulfides (SSP4 fluorescence) were examined in HEK 293 cells in standard incubator conditions (21% O_2_, 5% CO_2_, 74% N_2_) that are hyperoxic for these cells and under more physiological oxygen (physioxia) conditions (5% O_2_, 5% CO_2_, 90% N_2_) in gas permeable 96 well plates. Physioxia alone consistently increased H_2_S but had insignificant effects on polysulfides in otherwise untreated cells (Figure 1). All three MnPs at 3 μM decreased intracellular H_2_S and more than doubled polysulfides in both hyperoxic and physioxic environments and MnP-stimulated polysulfide production was greater in physioxic cells than it was in hyperoxic cells (Figure 1).

Similar results were observed with 0.3 μM MnPs (Appendix A) and with 1 μM MnTE (not shown). Polysulfides were still significantly increased, albeit less so, at 0.03 μM MnP although the effects on H_2_S were less pronounced (Appendix A).

Figure 2 shows the time- and concentration-dependent effects of MnTE on H_2_S (AzMC) and polysulfides (SSP4) in HEK 293 cells expressed as the percentage change from before to after MnTE treatment. The magnitude of all responses increased with MnTE concentration. However, the effects of MnTE on both H_2_S and polysulfides appeared to be independent of the oxygen status of the cells and relatively independent of the duration of MnTE exposure.

Figure 3 shows the effects of 24 h exposure to 1 μM MnPs on intracellular sulfur distribution in HEK293 cells. MnTE and MnHex significantly reduced cellular H_2_S and all three MnPs reduced H_2_S in organelles that have been identified in other studies as mitochondria [7].

All three MnPs also increased cellular polysulfides and this appeared to occur in nucleoli and what appeared to be fibrillary-like structures, although the specific intracellular locations where these occurred have not yet been identified. These results confirm that MnPs affect endogenous sulfur concentrations and suggest that the MnPs consume H_2_S in the production of polysulfides as they do in cell-free conditions [6].

### 2.2. Effects of MnPs on Sulfur Metabolism in Mesenchymal Stem, Bovine Pulmonary Artery Smooth Muscle, A549 and HT29 Cells

The effects of oxygen tension and 1 μM MnTE, MnBuOE and MnHex on human bone marrow derived stem cells are shown in Figure 4. H_2_S production was significantly (*p < 0.05*) greater in 5% (physioxic) O_2_ than in 21% (hyperoxic) O_2_ at all but 24 h, whereas the effects of O_2_ on polysulfides were less pronounced. All three MnPs decreased cellular H_2_S production and increased polysulfides.

H_2_S was also increased by the lower physioxic O_2_ tension in bovine pulmonary artery smooth muscle cells, adenocarcinomic human alveolar basal epithelial (A549) cells, and human colon cancer (HT29) cells, with minimal effects on cellular polysulfides (Figure 5). Consistent with the previous findings, 1 μm MnTE decreased H_2_S and increased polysulfides in all three cells and this appeared to be unaffected by O_2_ tension (Figure 5). Collectively, these results show that MnPs decrease intracellular H_2_S and increase intracellular polysulfides irrespective of cell type and oxygen status.

### 2.3. Effects of Ascorbic Acid on MnTE Oxidation of H_2_S in HEK and A549 Cells

The biological actions of MnPs are thought to be due to redox cycling where oxidized Mn(III)P is reduced by a reductant such as ascorbic acid (AA) and then re-oxidized by oxygen which generates superoxide [4,8,9,10]. The superoxide then spontaneously (or catalyzed by SOD or MnPs if present) dismutes to form peroxide and oxygen with the ultimate biological effects exerted by the peroxide. To determine if MnPs could redox cycle using O_2_ and AA to produce ROS that then would oxidize H_2_S, we treated HEK 293 and A549 cells with 1 μM MnTE and 1 mM AA and monitored H_2_S and polysulfide metabolism in 21% O_2_. As shown in Figure 6A,B, MnTE decreased H_2_S and increased polysulfides in both cell lines consistent with previous observations (cf. Figure 1 and Figure 5). AA alone did not affect H_2_S in HEK293 cells, but reduced H_2_S in A549 cells and increased polysulfides in both cell lines. AA plus MnTE decreased polysulfides compared to MnTE alone in HEK 293 cells but did not affect A549 cells. These results suggest that MnTE does not redox cycle with AA and O_2_ to form ROS that subsequently oxidizes H_2_S in either HEK293 or A549 cells.

### 2.4. MnP Metabolism of H_2_S in Buffer is Independent of Ascorbic Acid and H_2_O_2_

To determine if MnP generation of polysulfides from H_2_S requires AA, we evaluated the effects of 1 mM AA on the reaction between 1 μM MnPs and H_2_S in buffer in 21% O_2_. As shown in Figure 7A, polysulfide production was not increased by incubation of any MnP with AA in combination with either 100 or 300 μM H_2_S. We then incubated H_2_S with H_2_O_2_ in the presence or absence of MnTE (Figure 7B–D). H_2_O_2_ alone from 10 μM to 1 mM concentration-dependently increased polysulfide production in normoxia and hypoxia, albeit at a slow rate. MnTE increased polysulfide production, however, and this was concentration-dependently inhibited by 100 μM and 1 mM H_2_O_2_. These results suggest that H_2_O_2_ does not augment MnTE catalysis of H_2_S, and it appears to be inhibitory at higher concentrations [6].

## 3. Discussion

In the present studies, we show that MnPs produced polysulfides in a variety of cell lines and that this appeared to result from MnP oxidation of endogenously produced H_2_S. This hypothesis is also supported by our observation that MnPs decrease cellular H_2_S. In addition, the effects of MnPs on cellular H_2_S and polysulfides were concentration-dependent and could be observed at concentrations as low as 0.03 μM. This illustrates the efficacy of MnPs in cellular sulfur metabolism. Our studies also suggest that MnP-catalyzed H_2_S oxidation is efficient at relatively low levels of O_2_ and H_2_S.

### 3.1. Cellular Oxygen and Sulfur Availability

Reducing ambient oxygen from 21% to 5% increased intracellular H_2_S (AzMC fluorescence), consistent with previous studies showing that oxygen-dependent metabolism of H_2_S is a mechanism for short- and long-term cellular oxygen sensing [7,11]. H_2_S is constitutively produced from cysteine catabolism via the transsulfuration pathway and metabolized in the mitochondrion, initially by the mitochondrial enzyme sulfur:quinone oxidoreductase [12]. As H_2_S is oxidized by SQR, electrons are transferred to complex III via ubiquinone and shuttled down the electron transport chain (ETC) to ultimately reduce oxygen at complex IV. As oxygen levels fall, electron flux down the ETC decreases and as H_2_S oxidation can no longer keep pace with production, H_2_S increases. This mechanism inversely couples oxygen availability to H_2_S concentration.

A fall in oxygen can also increase H_2_S via other mechanisms. There is a large store of polysulfides in cells [13,14,15,16,17]. A fall in oxygen will increase the reducing environment in cells and reductants such as ascorbate and thioredoxin can then release H_2_S from polysulfides and thiosulfate [18,19,20,21]. As we did see a slight decrease in polysulfides when cells were in 5% oxygen (Figure 1), it is possible some H_2_S was derived from polysulfides. H_2_S also induces HIF-1α [22] and HIF-1α in turn increases expression of H_2_S-producing enzyme cystathione γ-lyase suggesting the potential for complex interplay between these hypoxia response systems that should be pursued in the future.

Our studies also suggest that oxygen availability is not a limiting factor on H_2_S oxidation. We have previously shown that MnP oxidation of H_2_S in buffer consumes O_2_, even at very low O_2_ tensions [6]. We also estimated that HEK293 cells in 21% O_2_ produce approximately 2–4 × 10^−16^ mol of H_2_S per hour per cell and this would require only 0.2–0.3% of the total O_2_ consumed by these cells [23]. Therefore, it is likely that there is sufficient O_2_ in cells, even at 5% O_2_, to support the additional oxidation of H_2_S by MnPs.

### 3.2. MnP Utilization of Intracellular Sulfur

It is evident that an increase in MnP concentration is associated with a decrease in cellular H_2_S production and an increase in polysulfide production (Figure 2). We propose that the decrease in H_2_S is the result of MnP-driven oxidation of H_2_S to polysulfides. It is less clear what the source of H_2_S is and where it comes from.

It is generally accepted that the concentration of H_2_S in cells or plasma is less than 1 μM as the human nose can detect as little as 0.5–1 μM H_2_S in solution [24] and there is no appreciable noxious odor of rotten eggs emanating from healthy animal cells or blood. However, the potential significance of H_2_S as a significant source of polysulfides stems not from the absolute concentration of H_2_S but rather the rate of H_2_S production relative to its metabolism and the extensive intracellular sulfur pool from which H_2_S can be derived.

HEK293 cells have an approximate cell diameter of nearly 14 μm [25] and a cell volume of 1.5 × 10^−9^ mL. If each cell produces 3 × 10^−16^ mol of H_2_S per cell [23], intracellular H_2_S could theoretically increase to 200 μM in one hour if H_2_S was not metabolized. Mitochondrial oxidation of H_2_S to sulfate normally keeps pace with H_2_S production but this can quickly change as evidenced by the rapid rise in H_2_S associated with hypoxia [11] and by the longer term responses observed in the present and other studies [7]. While it is tempting to propose that MnP-oxidation diverts H_2_S generated from cysteine catabolism away from mitochondrial oxidation, this does not appear to be the case. Figure 2 shows that while MnP-catalyzed polysulfide formation was observed in cells in both 21% and 5% O_2_, when polysulfide production was normalized to the percentage change in absolute fluorescence, the effect of MnPs on both AzMc and SSP4 fluorescence appeared to be independent of oxygen status. This suggests that H_2_S is not diverted from normal mitochondrial metabolism, nor dependent on hypoxia-increased cellular reductants or coupled to HIF-1α.

Recent evidence suggests that hydropersulfides (RSSH) and polysulfides (RSS_n_R’, where R and/or R’ are an alkyl or H and n > 1) are far more prevalent in cells than previously thought and that they exist in a dynamic equilibrium with H_2_S and other small inorganic thiols as well as cysteine (CysS_n_H), glutathione (GS_n_H) and protein (PS_n_H) per- and polysulfides [15,26,27,28,29,30]. Thus, there appears to be a large source/sink for H_2_S in cells that may be utilized by MnPs. The identity of this source and the mechanism by which H_2_S is mobilized from it, remains to be identified.

### 3.3. Antioxidant Function of MnP-Generated Polysulfides

A well known therapeutic attribute of MnPs is their ability to protect healthy cells and increase susceptibility of malignant cells to radiation therapy. This process has been attributed in part to increased H_2_O_2_ which, in a process catalyzed by MnP, oxidizes signaling proteins such as NFκΒ [4,10,31]. Our results suggest that persulfides produced by MnP oxidation of H_2_S could have a similar effect with the added bonus of direct oxidant scavenging by persulfides.

Polysulfides generated from MnP oxidation of H_2_S could contribute to both short- and long-term antioxidant effects. Short-term benefits are derived from the direct antioxidant effects of polysulfides. Counterintuitive as it may seem, oxidization of H_2_S, which is the most reduced form of sulfur (oxidation state −2) to per- and polysulfides (oxidation states −1 or 0) produces far better antioxidants than H_2_S [15,27,32]. We have also demonstrated this antioxidant potency with the garlic-derived polysulfides [33]. Addition of garlic oil (GO) or diallyl trisulfide (DATS), the active polysulfide in GO, increased antioxidant activity in HEK 293 cells and this was correlated with an increase in intracellular polysulfides, not an increase in H_2_S. The present studies show that much of the MnP-induced increase in cellular polysulfides occurs during the initial 4 h and this could provide a short-term benefit through the direct antioxidant activity of these molecules.

Long-term benefits of polysulfides are also derived from their modulation of antioxidant response mechanisms. Polysulfides increase intracellular antioxidant activity via persulfidation of Keap1 which causes Keap1 to dissociate from Nrf2, similar to the proposed mechanism of Keap-1 peroxidation [34,35,36,37,38,39]. Nrf2 then translocates to the nucleus where it activates the suite of cytoprotective antioxidant response elements [40,41]. Other cytoprotective mechanisms mediated by polysulfides can be found in recent reviews [26,28,42].

### 3.4. Mechanism of MnP-Catalyzed H_2_S Oxidation in Cells

Based on our studies in cell-free conditions, we recently proposed that polysulfides were produced by MnP-catalyzed redox cycling between H_2_S as the reductant and O_2_ as the oxidant [6]. In these reactions two molecules of Mn(III) are reduced by two hydrosulfide anions which produces two hydrosulfide radicals (Equation (1)). The two hydrosulfide radicals react with each other producing hydrogen persulfide (Equation (2)) and the reduced Mn(II) is re-oxidized by O_2_ forming Mn(III) and superoxide (O_2_^•−^; Equation (3)) and the latter is then spontaneously, or catalyzed by MnP, dismuted to hydrogen peroxide (H_2_O_2_) and water (Equation (4)).
2Mn^III^P^5+^ + 2HS^−^ → 2Mn^II^P^4+^ + 2HS^•^(1)
2HS^•^ → H_2_S_2_(2)
2Mn^II^P^4+^ + 2O_2_ → 2Mn^III^P^5+^ + 2O_2_^•−^(3)
2O_2_^•−^ + 2H^+^ → H_2_O_2_ + O_2_(4)

We also proposed that the hydrogen peroxide produced in Equation (4) can re-oxidize either Mn^II^P^4+^ or Mn^III^P^5+^ resulting in formation of high-valent MnP, Mn^IV^ (Equation (5)) or Mn^V^P (Equation (6)) and that these high-valent MnPs can then oxidize HS^−^ (Equations (7) and (8), respectively) producing additional hydrosulfide radicals and subsequently polysulfides.
Mn^II^P^4+^ + H_2_O_2_ → O = Mn^IV^P^4+^ + H_2_O(5)
Mn^III^P^5+^ + H_2_O_2_ → (O)_2_Mn^V^P^3+^ + 2H^+^(6)
O = Mn^IV^P^4+^ + HS^−^ + 2H^+^ ↔ Mn^III^P^5+^ + HS^•^ + H_2_O(7)
(O)_2_Mn^V^P^3+^ + HS^−^ + 2H^+^ ↔ O = Mn^IV^P^4+^ + HS^•^ + H_2_O(8)

However, we did not observe a consistent increase in MnP-catalyzed polysulfide production in cells in the presence of excess ascorbic acid or in buffer with either excess ascorbic acid or hydrogen peroxide, suggesting that neither ascorbate nor peroxide contribute to MnP-catalyzed polysulfide production in cells. This suggests that not only are the reactions given by Equations (1)–(4) the predominate reactions of MnP oxidation of H_2_S to polysulfides, but they also suggest that the cytoprotective effects of MnPs can be achieved through sulfur metabolism.

### 3.5. Efficacy of MnTE, MnTnHex and MnTnBuOE

The three MnPs appear to be nearly equally efficacious in producing polysulfides in cells as measured by SSP4 fluorescence, although the confocal studies suggested that MnTE was slightly less potent (Figure 3). Two factors can contribute to MnP activity in cells, direct catalytic efficiency and bioavailability. We showed in our previous study in cell-free conditions that MnTE was the most efficacious followed by MnTnBuOE and MnTnHeX [6]. The difference between MnP activity in buffer and cells appears to be due to bioavailability as MnTE accumulates in 4T1 cells to a lower extent than the more lipophilic MnTnHex [9].

### 3.6. Potential Limitations of the Study

#### 3.6.1. Specificity of Fluorophores

As with any study employing fluorophores, specificity is always a concern as it is virtually impossible to account for all potential interfering substances. AzMC and SSP4 specificity have been examined relative to other sulfur compounds as well as reactive oxygen and nitrogen species and both have been shown to be sufficiently specific for their respective analysis [23,43,44]. Clearly, there needs to be continual awareness of potential interfering substances with all fluorescent probes, however, we are reasonably confident that our probes are acceptable reporters of their respective RSS as employed in this study.

#### 3.6.2. Effects of Fluorophores on Sulfur Metabolism

AzMC, MeRho and SSP4 are irreversible fluorophores that were present throughout the entire experiment. In this way, they provide a historical record of sulfur metabolism over the entire experiment and the results are less sensitive to transient variations in the concentration of sulfur species and to the relatively slow response characteristics (tens of minutes) of the fluorophores. However, it is also recognized that when used this way they can also consume H_2_S and polysulfides and thereby affect sulfur metabolism. We do not know to what extent this could affect our results. However, our findings are consistent with our previous observations of MnP oxidation of H_2_S and polysulfide production in cell-free conditions [6], suggesting to us that our general conclusion that MnPs also produce polysulfides from H_2_S in cells is valid.

### 3.7. Summary

The biological activity of SOD-mimetic MnPs is often attributed to their involvement in ROS metabolism which may then affect cysteine thiols in glutathione and proteins. It is now evident that these compounds also oxidize H_2_S and form polysulfides, mainly H_2_S_2_. As H_2_S-driven catalysis appears to be able to function under relatively low oxygen levels, this would allow polysulfide formation to proceed in the relatively hypoxic environment of the mitochondrion, which is where MnPs preferentially accumulate and where it appears that much H_2_S is produced. Direct cytoprotective effects of MnP oxidation of H_2_S can be derived from the direct antioxidative properties of polysulfides as well as prolonged activation of Nrf2. The ability of MnPs to oxidize H_2_S in low O_2_ environments may take on even more therapeutic significance in hypoxic cells, such as solid tumors.

## 4. Materials and Methods

### 4.1. Chemicals

SSP4 (3′,6′-di(*O*-thiosalicyl)fluorescein) was purchased from Dojindo Molecular Technologies Inc. (Rockville, MD, USA). All other chemicals were purchased from either Sigma-Aldrich (St. Louis, MO, USA) or ThermoFisher Scientific (Grand Island, NY, USA).

Phosphate buffer (PBS; in mM): 137 NaCl, 2.7, KCl, 8 Na_2_HPO_4_, 2 NaH_2_PO_4_. pH was adjusted with 10 mM HCl or NaOH to 7.4. Please note that we use H_2_S to denote the total sulfide added (sum of H_2_S + HS^−^) usually derived from Na_2_S. Note: S^2−^, often thought of as part of the H_2_S + HS^−^ equilibrium, does not exist under these conditions [45].

### 4.2. Cells

Human embryonic kidney (HEK 293) cells, bovine pulmonary artery smooth muscle (BPASMC) cells, human bone marrow derived stem cells, adenocarcinomic human alveolar basal epithelial (A549) cells, and human colon cancer (HT29) cells were cultured and maintained at 37 °C in a 21% O_2_/5% CO_2_ humidified incubator supplemented with DMEM (low glucose) containing 10% FBS and 1% Pen/Strep. After trypsinization, cells were transferred from T-25 tissue culture flasks to gas-permeable 96-well plates (Coy Laboratory Products, Inc. grass Lake, MI, USA) and grown to >75% confluence. AzMC (7-azido-4-methylcoumarin) and SSP4 were used to monitor H_2_S and polysulfide production, respectively. The maximum MnP concentration used in these experiments was 3 μM, as previous studies have shown that above this concentration, the MnPs optically interfere with AzMC and SSP4 fluorescence [6].

Cells cultured under the above conditions are ‘hyperoxic’ relative to their ‘physioxic’ environment in vivo [46]. Due to the fact that exposure to more physioxic conditions also affects sulfur metabolism [7], a number of additional experiments were performed at more appropriate oxygen levels (5% O_2_/5% CO_2_/balance N_2_) in a model 856-HYPO hypoxia chamber (Plas Labs, Inc. Lansing, MI, USA) at 37 °C. They were covered with a standard 96-well plate cover before they were removed from the hypoxia chamber for plate reader measurements.

### 4.3. Confocal Microscopy

HEK 293 cells were cultured as above and transferred to 8 chamber well plates mounted on #1.5 German borosilicate coverglass (Fisher, Grand Island, NY, USA) and grown to ~75% confluence. They were then treated with either the H_2_S-specific fluorophore MeRho-Az (10 μM; [47]) or the polysulfide-specific fluorophore, SSP4 (20 μM; [48]) and then either no MnP or 1 μM of MnTE, MnTnBuOE or MnTnHex for 24 h in 21% O_2_/5% CO_2_ and examined with a Zeiss LSM 710 confocal microscope and 63X objective (1.4 NA) magnification. Imaging conditions were kept the same for all samples. Fluorescence intensity from a minimum of 15 images of whole cells of both MeRho-Az and SSP4 treated cells was quantified using Image J without any prior image adjustments; representative micrographs were uniformly brightened for clarity.

### 4.4. Fluorescence Measurements in Buffer

Compounds of interest were aliquoted into black 96 well plates in a darkened room and fluorescence was measured on the SpectraMax plate reader. Fluorescence was typically measured every 10 min over 90 min, although most reactions were completed within the first 10–20 min. Excitation/emission wavelengths for 3′,6′-di(*O*-thiosalicyl)fluorescein (SSP4), and 7-azido-4-methylcoumarin (AzMC) were 482/515 and 365/450 nm, respectively, per manufacture’s recommendations.

### 4.5. Calculations

Results are expressed as mean +SE. Statistical analysis was determined by one-way ANOVA with Holm-Sidak for multiple comparisons. Significance was assumed at *p ≤ 0.05.*

## Figures and Tables

**Figure 1 molecules-25-00980-f001:**
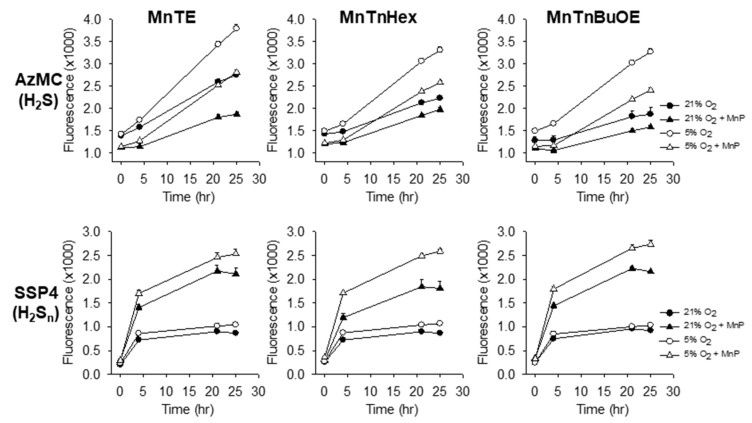
Effects of 3 μM MnPs on H_2_S (AzMC) and polysulfide (H_2_S_n_; SSP4) in HEK 293 cells in hyperoxia (21% O_2_, solid symbols) and physioxia (5% O_2_, open symbols). Intracellular H_2_S were increased in physioxia compared to hyperoxia (*p < 0.001*), whereas polysulfides were unaffected. All MnPs decreased intracellular H_2_S and increased intracellular polysulfides in both O_2_ environments (*p < 0.001*). Mean +SE, n = 16 wells per treatment, error bars may be covered by symbols.

**Figure 2 molecules-25-00980-f002:**
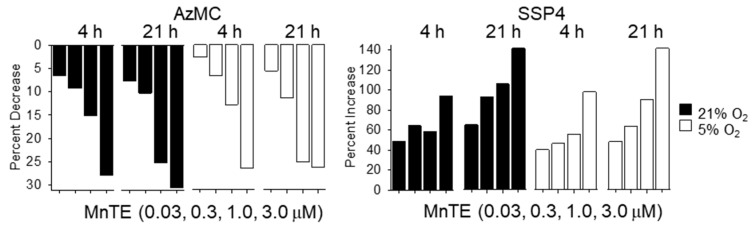
Concentration-dependent effects of MnTE on the percentage decrease in intracellular H_2_S and percentage increase in polysulfides (H_2_S_n_) after 4 and 21 h in hyperoxia (21% O_2_) or physioxia (5% O_2_). Values derived from mean values of Figs.1 (3 μM), S1 (0.3 μM) and S2 (0.03 μM) and 1.0 μM (not shown). Decreases in H_2_S and increases in polysulfides are dependent on MnTE concentration.

**Figure 3 molecules-25-00980-f003:**
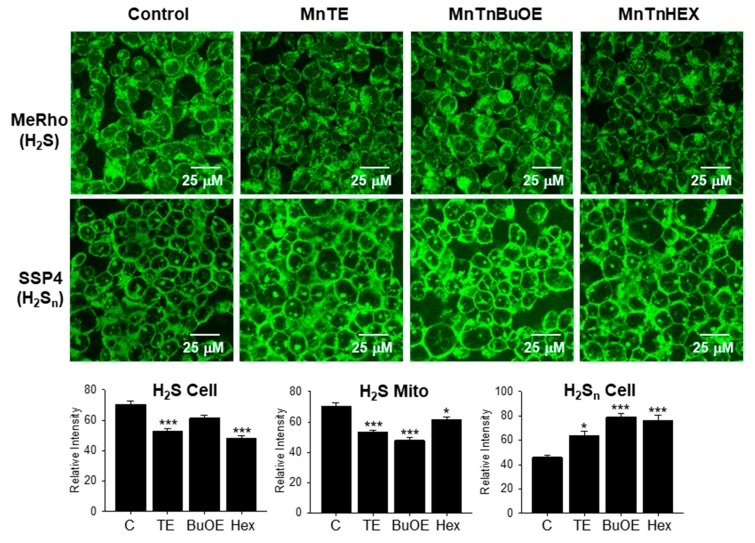
Confocal micrographs showing the effects of 1 μM MnPs on H_2_S (MeRho-Az fluorescence) and polysulfides (H_2_S_n_; SSP4 fluorescence) in HEK 293 cells. H_2_S fluorescence is reduced in MnTE and MnTnHex cells and in all MnP-treated mitochondria, whereas polysulfide fluorescence is greater in all MnP-treated cells. Micrographs uniformly enhanced for clarity, bar graphs from original, unaltered micrographs, mean + SE, n = 15 cells; *, *p < 0.05*; ***, *p < 0.001*.

**Figure 4 molecules-25-00980-f004:**
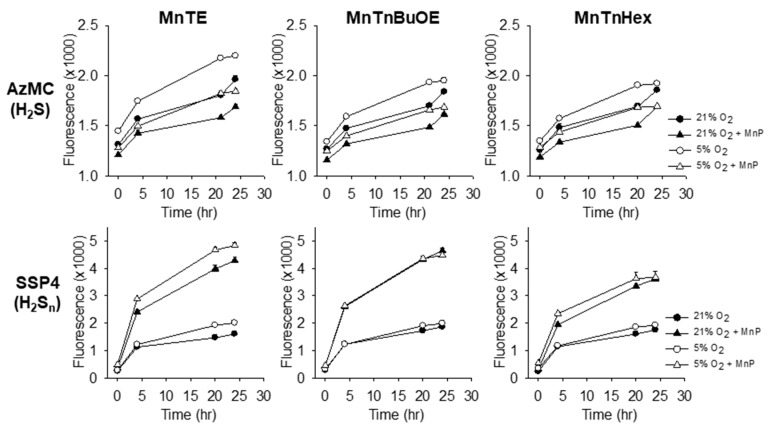
Effects of 1 μM MnPs on cellular H_2_S (AzMC) and polysulfide (H_2_S_n_; SSP4) in bone marrow-derived stem cells in hyperoxia (21% O_2_, solid symbols) and physioxia (5% O_2_, open symbols). Intracellular H_2_S was increased (*p < 0.01*) in physioxic cells at all but 24 h, whereas polysulfides were unaffected. All MnPs decreased intracellular H_2_S and increased intracellular polysulfides in both normoxia and hypoxia (*p < 0.001)*. Mean +SE, n = 8 wells per treatment, error bars may be covered by symbols.

**Figure 5 molecules-25-00980-f005:**
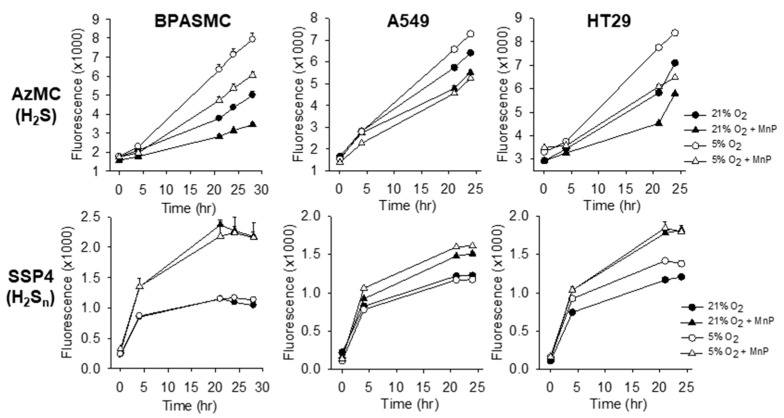
Effects of 1 μM MnTE on cellular H_2_S (AzMC) and polysulfide (H_2_S_n_; SSP4) in bovine pulmonary artery smooth muscle cells (BPMASMC), A549 and HT29 cells in hyperoxia (21% O_2_, solid symbols) and physioxia (5% O_2_, open symbols). Intracellular H_2_S was increased (*p < 0.01*) in physioxic cells, whereas polysulfides were unaffected by O_2_ tension. MnTE decreased cellular H_2_S and increased polysulfides in all cells (*p < 0.001*). Mean +SE, n = 8 wells per treatment, error bars may be covered by symbols.

**Figure 6 molecules-25-00980-f006:**
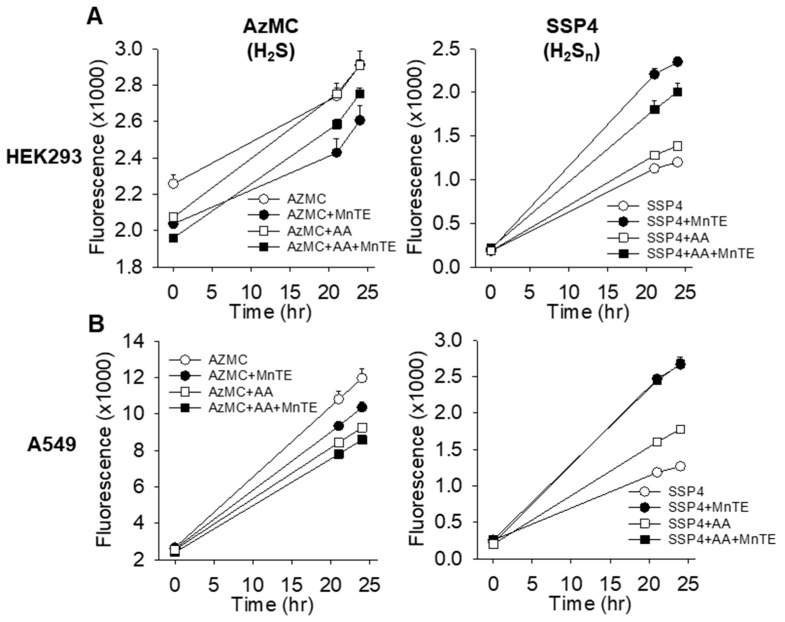
Effects of 1 mM ascorbic acid (AA) on H_2_S and polysulfides (H_2_S_n_) alone or in combination with 1 μM MnTE in HEK 293 (**A**) and A549 (**B**) cells. MnTE decreased H_2_S and increased polysulfides. AA alone did not affect H_2_S in HEK 549 cells but reduced H_2_S in A549 cells and increased polysulfides in both cell lines. AA plus MnTE decreased polysulfdes compared to MnTE alone in HEK 293 cells but did not affect A549 cells. Mean + SE, n = 8 wells per treatment.

**Figure 7 molecules-25-00980-f007:**
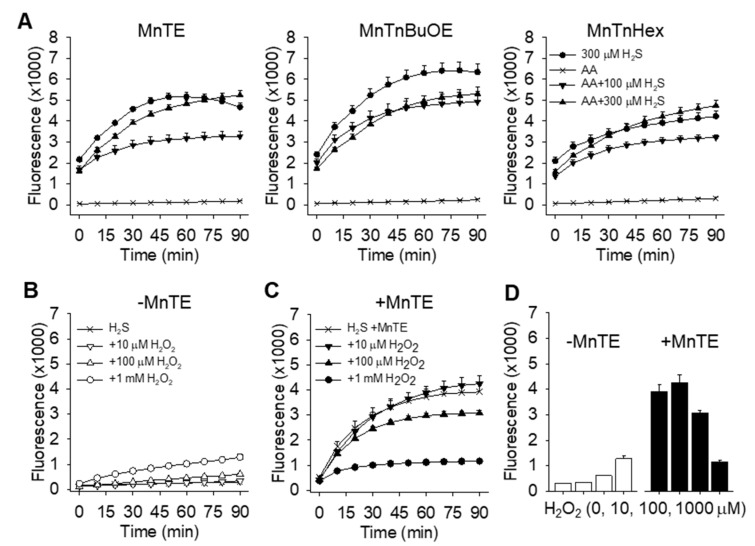
Effects of ascorbic acid (AA) and hydrogen peroxide (H_2_O_2_) on MnP formation of polysulfides (H_2_S_n_; SSP4 fluorescence) from H_2_S in 21% O_2_. (**A**) AA (1 mM) does not increase polysulfide production from 100 or 300 μM H_2_S by 1 μM MnPs. (**B**) H_2_O_2_ alone has minimal effect on polysulfide production from 300 μM H2S in the absence of MnTE, and only decreases polysulfide production in the presence of 1 μM MnTE (**C**). (**D**) Summary of the effects of H_2_O_2_ from **B** and **C** at 90 min. Mean + SE n = 4 wells per treatment.

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
