# Peer review of "Effects of Manganese Porphyrins on Cellular Sulfur Metabolism"

_molecules, 2020, doi:10.3390/molecules25040980_

Round 1

Reviewer 1 Report

The manuscript is interesting since clarifying the mechanisms that govern cell defense against oxidative stress, as well as the discovery of substances that may ameliorate oxidative stress, are clinically significant. In the submitted study, the authors suggest that manganese porphyrins may ameliorate oxidative stress by an immediate cascade of reactions, whereas the formatting polysulfides may activate Nrf2 protecting cells in the long term.

Comments

It would be helpful for the authors to explain better some of the described molecular mechanisms. For instance, in the sentence which notes that low oxygen enhances H2S production, they can add the known information that hypoxia through activation of HIF upregulates the expression of the H2S producing enzymes CBS, CSE, and 3-MST. Such explanations would increase the audience of the manuscript. The molecular mechanism involved in the decreased H2S production due to cell treatment with manganese porphyrins was not evaluated. Could it be solely the results of H2S conversion to H2S2 and other polysulfides? The authors should note their opinion. H2S, by altering Keap1, can activate Nrf2 directly. The authors claim that polysulfides also activate Nrf2 and attributed to the above that long term antioxidant features of manganese porphyrins. However, since the manganese porphyrins also decrease H2S, the polysulfides should be more potent nrf2 activators than H2S to support the authors’ claim. Is there any evidence for the latter? In the absence of it, this hypothesis is very weak. Extending the previous comment, it would be nice for the authors to show the effect of manganese porphyrins on Nrf2 activation, for instance, with WB. In the text that corresponds to figure 3, the authors make comments about the subcellular distribution of H2S and polysulfides. However, figure 3 magnification does not allow such statements. (Bar 25 μm, HEK-293 cell diameter 13 μm). Generally, the manuscript is written in a form that fits a reader with a strong chemical background. I suggest the authors expanding the text in a way that would make the reading and the understanding of the study easier for readers with a biological and/or medical background as well.

Author Response

Reviewer 1

Comments

It would be helpful for the authors to explain better some of the described molecular mechanisms. For instance, in the sentence which notes that low oxygen enhances H2S production, they can add the known information that hypoxia through activation of HIF upregulates the expression of the H2S producing enzymes CBS, CSE, and 3-MST. Such explanations would increase the audience of the manuscript.

* We know that H2S decreases HIF-1alpha but we’re not aware of studies showng that HIF-1alpha increases 3-MST.  We put in a bit on HIF-1alpha per the reviewer’s suggestion but we’re not sure what significance of this is to the story as hypoxia didn’t affect MnP oxidation of H2S to polysulfides.

 The molecular mechanism involved in the decreased H2S production due to cell treatment with manganese porphyrins was not evaluated. Could it be solely the results of H2S conversion to H2S2 and other polysulfides?

* Yes we believe this is the mechanism.  We clearly showed in our 2019 Antioxidants paper that MnPs directly oxidize H2S to H2S2 in buffer and I think provide a reasonably sound mechanism for this as discussed further in the present paper.

The authors should note their opinion. H2S, by altering Keap1, can activate Nrf2 directly. The authors claim that polysulfides also activate Nrf2 and attributed to the above that long term antioxidant features of manganese porphyrins. However, since the manganese porphyrins also decrease H2S, the polysulfides should be more potent nrf2 activators than H2S to support the authors’ claim. Is there any evidence for the latter? In the absence of it, this hypothesis is very weak.

* Our apologies for the apparent confusion.  We did not mean to infer that H2S activates Keap-1and we agree with the reviewer that the MnPs oxidize H2S to polysulfides.  It is the persulfides that persulfidate Keap-1 thereby freeing Nrf2 and allowing it to translocate to the nucleus.  Persulfidation of Keap-1 has been well demonstrated in the literature and these points have been clarified in the discussion.

 Extending the previous comment, it would be nice for the authors to show the effect of manganese porphyrins on Nrf2 activation, for instance, with WB.

* I do not have the resources for this and ask the reviewer’s indulgence.  I also believe it is beyond the scope of the present study and there are a number of papers showing the effect of polysulfides on this.

In the text that corresponds to figure 3, the authors make comments about the subcellular distribution of H2S and polysulfides. However, figure 3 magnification does not allow such statements. (Bar 25 µm, HEK-293 cell diameter 13 µm).

* Our apologies for the confusion.  The micrographs were enlarged when the measurements were made.

Generally, the manuscript is written in a form that fits a reader with a strong chemical background. I suggest the authors expanding the text in a way that would make the reading and the understanding of the study easier for readers with a biological and/or medical background as well.                            

* More biology has been added in the introduction and discussion.  We hope that this is sufficient.

Reviewer 2 Report

Major points:

In the Supplemental Figures it has not been indicated the meaning of the symbols (triangles and circles).

Despite the authors show clear concentration-response relationships when the responses are expressed as percentage of H2S decrease or PS increase, these relationships are not very clear when looking at the fluorescence time-course curves for the MnPs at the different concentrations. It seems that there are no significant differences for the MnPs applied between 0.03 and 3 uM on the HEK cells. Why did the authors use 3 uM MnPs when these higher concentrations could lead to colateral effects? (Just remembering that the plasma concentrations achieved with these compounds is in the range 10 – 200 nM or below 1 uM when 10 mg/kg doses are orally or i.p. administered to humans).

Why do the cells produce more H2S under 5% O2 in comparison with 21% O2 but PS production is unaffected by the different O2 concentrations? How do the different O2 tensions affect the viability of the cells? Have the authors correlated these aspects? How do their findings on the effects of ascorbic acid apply to these responses?

It is not clear to which extent the cells can uptake each of the different MnPs (mainly due to their highly cationic nature and their different chemical structures and lipophilicities) or whether their intracellular effects are mediated by membrane receptors. The authors should at least try to explain why their effects on intracellular H2S/PS production are so similar in spite of their important chemical differences.

The authors should validate the intracellular and buffer fluorescence measurements of H2S and PS under inhibition of the producing enzymes and quenching of H2S, respectively. It is actually intriguing how they obtained so similar fluorescence intensity values when they studied intracellular PS and PS formation from H2S in the cell-free systems (in minutes!).

For additional information on this point, please refer to the article from Takanao et al. (2017; available at https://doi.org/10.1089/ars.2017.7070).

In the Discussion, the authors refer to the theoretical intracellular concentrations of H2S based on the cell volume and H2S-production (including mitochondria-derived H2S. However, mitochondria also use H2S as an electron donor via the enzyme sulfurquinone reductase (which leads to enhanced ATP and decreased O2- production). The authors should discuss the potential interference of the MnPs on the mitochondrial electron transfer chain.

In order to confirm the validity of the reactions in the cell systems, the authors should at least measure intracellular H2O2 concentrations in response to the MnPs. Otherwise, the exposure and discussion of these reactions do not contribute to the objectives of the manuscript.

Author Response

Major points:
1. In the Supplemental Figures it has not been indicated the meaning of the symbols
(triangles and circles).
*This is now so indicated both in the figures and in the legend.
2. Despite the authors show clear concentration-response relationships when the responses
are expressed as percentage of H2S decrease or PS increase, these relationships are not
very clear when looking at the fluorescence time-course curves for the MnPs at the different
concentrations. It seems that there are no significant differences for the MnPs applied
between 0.03 and 3 uM on the HEK cells. Why did the authors use 3 uM MnPs when these
higher concentrations could lead to colateral effects? (Just remembering that the plasma
concentrations achieved with these compounds is in the range 10 – 200 nM or below 1 uM
when 10 mg/kg doses are orally or i.p. administered to humans).
* We agree with the reviewer that the differences between 0.03 and 3.0 are relatively small (although they are significant, mostly at p<0.001) compared to the effects of 0.03 alone. We used 3 uM in most of our studies because this gave good responses in our previous cell-free experiments and we thought it would be the best here as well. In retrospect we probably should have used even lower concentrations, but limited resources prevent us from additional studies. However, we feel that this does not detract from the major findings of the study; MnPs affect sulfur metabolism in cells that are consistent with their effects in cell-free conditions.
3. Why do the cells produce more H2S under 5% O2 in comparison with 21% O2 but PS
production is unaffected by the different O2 concentrations? How do the different O2
tensions affect the viability of the cells? Have the authors correlated these aspects? How do
their findings on the effects of ascorbic acid apply to these responses?
* The effects of 5% vs 21% O2 on sulfur metabolism in cells including those in this study have been examined in detail and reported previously. Reference to this is in the discussion and we expanded on this for better clarity. Although 21% O2 is used extensively in cell culture (and we used it here for comparison) it is actually hyperoxic for all cells except very upper airways and the cornea. All the cells used in this study thrive at 5% O2 and produce more H2S. The effects of 5% O2 on polysulfides is less obvious because there is a large endogenous pool of polysulfides that often masks the O2 effect. This makes the effect of MnPs on polysulfides even more striking. The intracellular environment becomes more reduced at 5% O2 which should amplify the effects of exogenous ascorbate if this was a factor in polysulfide production. We did not see this and concluded that this was not a significant factor.
4. It is not clear to which extent the cells can uptake each of the different MnPs (mainly due
to their highly cationic nature and their different chemical structures and lipophilicities) or
whether their intracellular effects are mediated by membrane receptors. The authors
should at least try to explain why their effects on intracellular H2S/PS production are so
similar in spite of their important chemical differences.
* Good point! Although MnTE is more efficacious in buffer it is not as well taken up by cells as the other MnPs are more lipophilic and better acculmulated. This somewhat offsets the kinetic differences. This has been added to the discussion.
5. The authors should validate the intracellular and buffer fluorescence measurements of
H2S and PS under inhibition of the producing enzymes and quenching of H2S, respectively. It
is actually intriguing how they obtained so similar fluorescence intensity values when they
studied intracellular PS and PS formation from H2S in the cell-free systems (in minutes!).
For additional information on this point, please refer to the article from Takanao et al.
(2017; available at https://doi.org/10.1089/ars.2017.7070).
* Unfortunately that doi did not work, nor did a literature search of Takanao, 2017 and ARS, so we are not sure what article the reviewer is referring to. That said, we have examined H2S and polysulfide production in cells with inhibitors of CSE, CBS and 3-MST alone and in combination and we were never able to completely block either. In fact, the best we could do was ~50%. This is not surprising. I (Olson) have reviewed the ‘conventional’ and ‘unconventional’ pathways of sulfur metabolism (Olson, Biochem Pharmacol
149:77-90, 2018) and showed that there are numerous pathways that would not be expected to be affected by these inhibitors.
6. In the Discussion, the authors refer to the theoretical intracellular concentrations of H2S
based on the cell volume and H2S-production (including mitochondria-derived H2S.
However, mitochondria also use H2S as an electron donor via the enzyme sulfurquinone
reductase (which leads to enhanced ATP and decreased O2- production). The authors
should discuss the potential interference of the MnPs on the mitochondrial electron
transfer chain.
* We doubt that there is much of an effect. We have calculated that only 0.1-0.2% of cellular O2 consumption is needed to metabolize H2S (Olson., et al., FRBM, 146:139-149, 2019), hence reducing H2S would not significantly affect either O2 consumption or ATP production.
7. In order to confirm the validity of the reactions in the cell systems, the authors should at
least measure intracellular H2O2 concentrations in response to the MnPs. Otherwise, the
exposure and discussion of these reactions do not contribute to the objectives of the
* This is a good idea but we don’t know how to do it and be sure we are measuring H2O2 and not sulfur compounds. We have previously shown that redox-sensitive green fluorescent protein (roGFP; arguably the gold standard for ROS) is actually 200 times more sensitive to sulfur species (SS) than it is to ROS, that ROS, peroxide electrodes are 25 times more sensitive to SS and that DCF, MitoSox Red and Amplex Red also respond to SS; DCF is also readily oxidized by catalase further complicating the issue (DeLeon et al., Am.J.physiol. 310: R549-560, 2016). In order to try other methods we would have to also validate their selectivity for ROS compared to H2S and polysulfides This is considerably beyond our reach in terms of time and financial resources (especially the latter).

Reviewer 3 Report

The authors evidence that MnP effects should be also considered from the point of view of sulfur metabolism and not exclusively from that of Oxygen. Consequently our current understanding of MnPs effect will be enriched (complicated) by this contribution. In this respect this article is important.

I will thereafter rise remarks, which the authors should consider for improvement.

1) Probes :

I did not find in the manuscript references with regard to the specificity towards H2S and polysulfide of the probes used, could the authors provide some?

2) Kinetics (this applies to the interpretation of figures 1, 4, 5 , 6) :

In these figures is presented an increase in AzMC or SSP4 fluorescence over time. It is not completely clear how it is measured, are the probes present from the beginning and increase in fluorescence over time recorded in the same samples? or is there sampling at the different time with probe addition and measurement afterwards? One or the other would alter interpretation of these graphs. Probes be always present would withdraw some H2S/polysulfides from the "normal" cellular reaction scheme and thus report the production rate, not concentration, a risk is the possibility that changes reflect alteration of the reactivity of these probes by MnPs. In contrast separate dosage over time would probe concentration in the sample and reflect accumulation of H2S/polysulfides in cells/medium.

3) Effect of ascorbic acid § 2.3:

My understanding is that the key issue is that it was never observed that presence of AA significantly increased the generation rate/accumulation of polysulfides. If this interpretation is correct, I suggest to restrict the presentation of these results as such. I think that the authors make too much comments on modest or hardly interpretable variations when AA is alone while ordinate show significant variation in absolute values and relative increase. Moreover the legend indicates "Mean±SE n=8 wells per treatment" this looks like the result of a single experiment with 8 wells for each condition hence error bars would refer to the reproducibility within one experiment and not to independent experiments.

4) Discussion H2S production pathways and polysulfide formation :

lines 213-217 " While it is tempting to propose that MnP oxidation diverts... This suggest other pathways of H2S production may be utilized by MnPs to produce polysulfides". This relates to the issue of whether there is one pool of H2S subject to the different consumption pathways or several pools with distinct fates. This may apply also to the reaction with the probe... Another point is confusing here as the authors tend to neglect (and make the reader even more to do so) the difference/dissotiation between concentration and flux. The increase in H2S observed in figure 1 relates an higher availability of H2S to the AzMC probe, depending on the experimental scheme (see 2 above) it may have slightly different meaning. Nevertheless as stated line 210 the largest part of H2S is consumed in other words while flux is large concentration remains low. This difference explains easily how H2S concentration could increase abruptly if consumption is impaired. The competition between polysulfide formation and mitochondrial oxidation would be dependent on the relative affinities of both pathways for H2S. If polysulfide formation shows high affinity for sulfide it could proceed at a rate independent from mitochondrial oxidation, which would consume "the rest".  

5) Reaction schemes (§ 3.3):

From lines 264-267 one concludes that tthe authors would rather consider reactions 1-4 to be involved while reaction 5-8 would be less likely to take place. Is this the correct interpretation? and if so perhaps should it be stated more clearly.

6) Other:

Abstract, line 36: spelling "observations" not "obsrvations".

Figure 2: no error bars ?

Position of titles in figures is not optimal (figures 6 and 7), consider that this may be an effect of conversion into PDF aggravated by local printer.

Figure 6: AzMC(H2S) and SSP4(H2Sn) too high and left drifted; A and B too high (B faces the ordinate of the "A" graph...) furthermore they should appear in larger characters than the cell type (or be withdrawn and replaced by top/bottom in the legend).

Figure 7: the titles MnTE, MnTnHex, MnTnBuOE are half cuttted (probably hidden by a white filling of the graph). 

Author Response

Reviewer 2

Comments and Suggestions for Authors

The authors evidence that MnP effects should be also considered from the point of view of sulfur metabolism and not exclusively from that of Oxygen. Consequently our current understanding of MnPs effect will be enriched (complicated) by this contribution. In this respect this article is important.

I will thereafter rise remarks, which the authors should consider for improvement.

1) Probes :

I did not find in the manuscript references with regard to the specificity towards H2S and polysulfide of the probes used, could the authors provide some?

* This has been added to the discussion 3.6.1. Specificity of fluorophores’ as part of the new general section ‘3.6.  Potential limitations of the study’

2) Kinetics (this applies to the interpretation of figures 1, 4, 5 , 6) :

In these figures is presented an increase in AzMC or SSP4 fluorescence over time. It is not completely clear how it is measured, are the probes present from the beginning and increase in fluorescence over time recorded in the same samples? or is there sampling at the different time with probe addition and measurement afterwards? One or the other would alter interpretation of these graphs.

* The probes were continuously present.  The pros and cons of this are now described in the section 3.6.2. ‘Effects of fluorophores on sulfur metabolism’.

Probes be always present would withdraw some H2S/polysulfides from the "normal" cellular reaction scheme and thus report the production rate, not concentration, a risk is the possibility that changes reflect alteration of the reactivity of these probes by MnPs. In contrast separate dosage over time would probe concentration in the sample and reflect accumulation of H2S/polysulfides in cells/medium.

* Section 3.6.2. ‘Effects of fluorophores on sulfur metabolism’ has been added to address sulfur consumption by fluorophores.  Clearly this is a concern and we hope it is satisfactorily addressed.

3) Effect of ascorbic acid § 2.3:

My understanding is that the key issue is that it was never observed that presence of AA significantly increased the generation rate/accumulation of polysulfides. If this interpretation is correct, I suggest to restrict the presentation of these results as such. I think that the authors make too much comments on modest or hardly interpretable variations when AA is alone while ordinate show significant variation in absolute values and relative increase.

* One of the main hypotheses of the biological actions of MnPs is the redox cycling of MnPs with oxygen and ascorbate to produce superoxide.  The superoxide then dismutes to peroxide which is toxic to malignant cells and cytoprotective to healthy cells.  Our results show that these same attributes can be derived from MnP oxidation of H2S to polysulfides and that this is independent of peroxide production.  This provides a novel mechanism for the action of these compounds and we feel that it is important to keep this information in the study.

Moreover the legend indicates "Mean±SE n=8 wells per treatment" this looks like the result of a single experiment with 8 wells for each condition hence error bars would refer to the reproducibility within one experiment and not to independent experiments.

* This is correct.  However, the consistency of our observations between all the different experiments, and even with different cells, suggests that is not a problem.

4) Discussion H2S production pathways and polysulfide formation :

lines 213-217 " While it is tempting to propose that MnP oxidation diverts... This suggest other pathways of H2S production may be utilized by MnPs to produce polysulfides". This relates to the issue of whether there is one pool of H2S subject to the different consumption pathways or several pools with distinct fates. This may apply also to the reaction with the probe... Another point is confusing here as the authors tend to neglect (and make the reader even more to do so) the difference/dissotiation between concentration and flux. The increase in H2S observed in figure 1 relates an higher availability of H2S to the AzMC probe, depending on the experimental scheme (see 2 above) it may have slightly different meaning. Nevertheless as stated line 210 the largest part of H2S is consumed in other words while flux is large concentration remains low. This difference explains easily how H2S concentration could increase abruptly if consumption is impaired. The competition between polysulfide formation and mitochondrial oxidation would be dependent on the relative affinities of both pathways for H2S. If polysulfide formation shows high affinity for sulfide it could proceed at a rate independent from mitochondrial oxidation, which would consume "the rest". 

* We agree.  Clearly this is a complex issue and one that we can’t adequately address in this study.  The continuous presence of fluorophores allows us to get an idea of the overall relative changes from one time point to the next, but we can’t discriminate between compartments or fluxes between them.  That said, the decrease in H2S production seems to correlate with the increase in polysulfide production and because this is consistent with out previous findings in cell-free conditions, it appears to offer a mechanistic explanation.

5) Reaction schemes (§ 3.3):

From lines 264-267 one concludes that tthe authors would rather consider reactions 1-4 to be involved while reaction 5-8 would be less likely to take place. Is this the correct interpretation? and if so perhaps should it be stated more clearly.

* It is correct.  This has been rewritten to that effect.

6) Other:

Abstract, line 36: spelling "observations" not "obsrvations".

* The abstract has been redone.

Figure 2: no error bars ?

* These were obtained from mean values so there were no error bars.  This is now indicated in caption.

Position of titles in figures is not optimal (figures 6 and 7), consider that this may be an effect of conversion into PDF aggravated by local printer.

* They appeared OK when submitted (see also comments below).  Additional care will be taken in the revision.

Figure 6: AzMC(H2S) and SSP4(H2Sn) too high and left drifted; A and B too high (B faces the ordinate of the "A" graph...) furthermore they should appear in larger characters than the cell type (or be withdrawn and replaced by top/bottom in the legend).

* See above.

Figure 7: the titles MnTE, MnTnHex, MnTnBuOE are half cuttted (probably hidden by a white filling of the graph).

* See above.

Round 2

Reviewer 1 Report

The authors improved the manuscript significantly. However, two points require further improvement.

  1. Regarding the effect of hypoxia and HIFs on H2S producing enzymes, there are data from the literature showing an increase of the above enzymes. This information is essential, may explain some of the results and the authors should include it in the second revision. Since the authors are not aware of, I add some of the related literature.

Takano N, Peng Y-J, Kumar Ganesh K, et al: Hypoxia-inducible factors regulate human and rat cystathionine β-synthase gene expression. Biochemical Journal 458: 203-211, 2014.

Wang M, Guo Z and Wang S: Regulation of Cystathionine γ-Lyase in Mammalian Cells by Hypoxia. Biochemical Genetics 52: 29-37, 2013.

Li M, Nie L, Hu Y, et al: Chronic intermittent hypoxia promotes expression of 3-mercaptopyruvate sulfurtransferase in adult rat medulla oblongata. Autonomic Neuroscience 179: 84-89, 2013.

  1. The authors repeatedly referred to the possible implication of Nrf2 in the detected results. Thus, showing the effect of manganese porphyrins on Nrf2, at least at one of the used cell lines, is pivotal, confirmatory of their claims, and also will show that in the specific experimental model, with the used concentrations of the reagents this system works as expected.

Reviewer 2 Report

The raised points have been answered and the present version of the manuscript has been significantly improved.